# Uncertainties in Electric Circuit Analysis of Anisotropic Electrical Conductivity and Piezoresistivity of Carbon Nanotube Nanocomposites

**DOI:** 10.3390/polym14224794

**Published:** 2022-11-08

**Authors:** Stepan V. Lomov, Nikita A. Gudkov, Sergey G. Abaimov

**Affiliations:** Center for Petroleum Science and Engineering, Skolkovo Institute of Science and Technology, 121205 Moscow, Russia

**Keywords:** carbon nanotubes, electrical conductivity, intrinsic conductivity, tunneling conductivity, piezoresistivity, gauge factor

## Abstract

Electrical conductivity and piezoresistivity of carbon nanotube (CNT) nanocomposites are analyzed by nodal analysis for aligned and random CNT networks dependent on the intrinsic CNT conductivity and tunneling barrier values. In the literature, these parameters are assigned with significant uncertainty; often, the intrinsic resistivity is neglected. We analyze the variability of homogenized conductivity, its sensitivity to deformation, and the validity of the assumption of zero intrinsic resistivity. A fast algorithm for simulation of a gauge factor is proposed. The modelling shows: (1) the uncertainty of homogenization caused by the uncertainty in CNT electrical properties is higher than the uncertainty, caused by the nanocomposite randomness; (2) for defect-prone nanotubes (intrinsic conductivity ~10^4^ S/m), the influence of tunneling barrier energy on both the homogenized conductivity and gauge factor is weak, but it becomes stronger for CNTs with higher intrinsic conductivity; (3) the assumption of infinite intrinsic conductivity (defect-free nanotubes) has strong influence on the homogenized conductivity.

## 1. Introduction

Prediction of the electrical properties of nanocomposites, in particular with carbon nanotubes (CNTs) as a filler, includes simulations of homogenized conductivity and piezoresistivity (change in the homogenized conductivity with deformation of nanocomposite), given the volume fraction and architecture of CNTs and their electrical properties. Nodal analysis of the conductive resistor network formed by CNTs above the percolation threshold in an insulative matrix is a well-established method for the simulation of the homogenized electrical conductivity of CNT-filled polymers. The resistor networks model [1] was applied to CNT nanocomposites in early 2000 [2,3,4]. In the next decades up to the present, these models were further developed in numerous publications, mostly based on random generation of a CNT assembly with its subsequent transformation into a resistor network (to name a few, [5,6,7,8,9,10,11,12,13,14,15]) or with alternative formulations of charge flow [16] and micromechanics-type calculations of homogenized conductivity [17,18,19]. The resistor network approach was further used as a basis for models of piezoresistive behavior for CNT nanocomposites [20,21,22,23,24,25,26,27]. Availability of such models opened a way for multi-scale simulations of CNT nanocomposites [28] and the optimization of sensing simulations based on complex machine learning algorithms [29], in line with micromechanical toughness optimization methods [30,31].

As in any simulation, the adequacy of any property prediction depends on the adequacy of input data. For the conductivity of CNT nanocomposites at a given volume fraction of the filler, the input includes (1) geometrical characteristics of CNTs (single- or multiwalled and the wall count, outer diameter, and length distribution), (2) their waviness, (3) geometrical characteristics of the CNT assembly (orientation distribution, and level of agglomeration), and finally, (4) the electrical characteristics of CNTs and their contacts. Apart from the number of walls and the CNT diameter, all these parameters for a given material are neither readily available from manufacturer’s datasheets, nor easily measurable, nor, at least, fully evaluated. Therefore, input data create a very considerable uncertainty, which should be accounted for in the results of calculations (i.e., propagation of error).

The present paper is dedicated to the electrical characteristics of CNTs and their contacts. It analyzes the uncertainty of the choice of parameters, including the intrinsic conductivity of CNTs themselves (up to an assumption of zero resistance of CNTs, the so-called ballistic limit) and tunneling conductivity of their contacts, plus the uncertainty of the following nodal analysis. This complex uncertainty has not been given proper attention in the literature, including works cited above; the authors provide reasons for their choice of a certain dataset but very rarely discuss possible deviations from the chosen values. Study [32] conducted an analysis of prediction variations in a simplified model of randomly oriented CNT networks depending on the chosen values of the intrinsic and tunneling conductivity. Recently published research [33] presents a statistical analysis of electro-mechanical properties of CNT nanocomposites, including conductivity and piezoresistivity, using micromechanical modelling of parallel arrays of CNTs for a wide range of tunneling energy barrier values and for a fixed value of the intrinsic CNT conductivity.

We will analyze (1) the range of variations for the conductivity tensor and its deformation sensitivity resulting from the nodal analysis and caused by the uncertainty of parameters, namely the intrinsic and tunneling CNT conductivities, and (2) the credibility limits for the assumption of the zero CNT intrinsic resistivity for two cases of the nanocomposite internal structure: isotropically oriented assembly of CNTs, resulting in an isotropic conductivity tensor, and aligned CNTs, leading to anisotropic conductivity.

Numerical simulations in this work are accomplished using Matlab version R2020a.

## 2. Uncertainty in Electrical Conductivity as Input Data

### 2.1. Intrinsic CNT Conductivity

#### 2.1.1. Physical Phenomena

Theoretically, the resistances of an ideal single-walled CNT (SWCNT) and a multi-walled CNT (MWCNT) are quantum (the so-called ballistic electron transport) which do not depend on the length of the conductive segment if this length is smaller than the mean free path of electrons. The latter is theoretically estimated as ~1 µm for SWCNTs and tens of micrometers for MWCNTs [34]. The typical distance *b* between CNT contacts along a CNT in a random CNT assembly can be estimated as $b=d/2V_{f}$ [35,36], where *d* is the CNT diameter, and *V_f_* is the volume fraction. For typical values *d* ~ 0.01 µm and *V_f_* ~ 0.01, this equation gives *b* ~ 0.5 µm, which is shorter than 1 µm of the free path. This suggests that the conductivity along CNT segments between two contact points should be taken as ballistic, and there exist experimental data supporting this statement [37,38,39]. However, the reliability of these measurements has been strongly criticized. Studies [40,41] argued that the used liquid–metal contact method gives “false positive” results for the ballistic nature of conductance. The theory of quantum resistance assumes the absence of the CNT structural defects. This assumption, although better justified on nano-scale than on macro-scale, still oversimplifies the description of the reality since CNT defects are experimentally observed and well-studied [42,43,44]. The presence of defects causes electron scattering and, thereby, the degradation of their mean free path length. For example, in [44] the mean free path length in MWCNTs was measured to be just 4.7 nm, which is a thousand times shorter than the theoretically suggested value for the ideal defect-free case. The authors of [44] suggested that the presence of defects cause a diffusive, Ohmic type of resistance, linearly increasing with the increase in the length of a conducting segment.

#### 2.1.2. Implementation in the Nodal Analysis

In the nodal analysis of CNT networks, there exist two approaches (see the references below in this section). The first assumes the Ohmic proportionality of the resistance of a CNT segment to its length, taking both intrinsic and tunneling resistances into account. The second approach assumes infinite conductivity along the CNTs, constructing a network of only the inter-CNT contacts.

With the first approach, conductance *G_CNT_* of a CNT segment of length *l_seg_* is calculated as:(1)$$G_{CNT}=g_{intr}\frac{A}{l_{seg}}$$
where *A* is the CNT cross-sectional area, and *g_intr_* is the volumetric conductivity of the CNT. The area is most often taken as the full area of the cross section, $A=\pi d^{2}/4$, where *d* is the outer CNT diameter. This definition is also often used when processing experimental data. In several experimental and modelling works, the presence of the internal hole is accounted for in calculations of the area *A*. Using the empirical link between the CNT diameter and the number of walls *N_w_* = (*d*[*nm*]−1) [45], the difference between the tubular and the full cross-section area can be estimated; it is in the range 10–20% for 5 < *d <* 35 nm. In the following, the full area of the cross section will be used; hence, *g_intr_* has the meaning of the effective volumetric conductivity; if a literature source relied on the tubular cross section, the data were re-calculated.

Figure 1a shows the results of measurements of *g_intr_* reported by [46] (measurement on single SWCNTs), [47,48] (single MWCNTs), and [44,49,50] (MWCNTs, aligned assembly). Even in the same series of experiments, the difference in *g_intr_* can be as large as 50 times. There exists a weak negative correlation between *g_intr_* and the CNT diameter (experimental data in Figure 1a); one can speculate that this is the result of a higher defect content in larger diameter CNTs. The CNT manufacturers do not provide data on CNT conductivity in their datasheets; an exception is MWCNTs NC7000, where the value, shown in Figure 1a (“internal test method” is referenced) is given in [51].

The popular choices of *g_intr_* for analyses of CNT networks are round values 10^3^ S/m [33], 10^4^ S/m [3,4,6,7,8,9,22], and 10^6^ S/m [10,15]. In these sources, no detailed arguments were given for the choice of this parameter, apart from a reference to the general range of it (through three decimal orders of magnitude). Studies of the influence of this choice have not been conducted, apart from the work [32], where the range 10^3^–10^6^ S/m is explored.

The reality is even more complex. Any individual CNT will have conductivity affected by chirality and defects, creating a distribution of *g_intr_*. All of these distributions are unquantified, to the best of authors’ knowledge.

As mentioned above, in a number of modelling studies [12,24,26,27,52,53,54,55], the conductance of CNT segments in a network was taken to be infinite, with the network conductance defined by the contacts (tunneling currents). The differences in homogenized conductivity calculated with this assumption and with finite conductivity of CNTs were not assessed in detail.

Based on the review in this subsection, we will investigate differences in the nodal analysis of a CNT network for the levels of the effective CNT conductivity between 10^4^ S/m and 10^6^ S/m. The level of 10^8^ S/m will be taken as very high, producing almost the same results as an assumption of infinite conductivity.

### 2.2. Tunneling Conductance of CNT Contacts

#### 2.2.1. Simmons’ Formula

The contact conductance between CNTs is governed by the fluctuation-induced tunneling electron transport [56] and depends on the probability of electrons to overcome the electric potential barrier between CNTs. The tunneling conductance can be described with Simmons’ formula [57], which for low voltage on a contact is [58]:(2)$$G_{tunn}=G_{0}\frac{\tau }{s}\frac{d^{2}}{32}\exp\left(-\tau s\right)$$
where *s* is the distance between the CNT surfaces, *s* ≥ *s_min_*, *d* is the CNT outer diameter, $G_{0}=2e^{2}/h$ = 7.722⋅10^−5^ S (*e =* 1.602⋅10^−19^ C is electron’s charge, *h =* 6.626⋅10^−34^ J⋅s is Plank’s constant), and
(3)$$\tau =\frac{4\pi \sqrt{2m\Delta E}}{h}$$
where *m* = 9.109⋅10^−31^ kg is the electron’s mass, and Δ*E* is the potential barrier.

Application of the low voltage variant of Simmons’ formula is justified by the fact that the dependence of tunneling conductivity on contact voltage is felt only if the voltage applied to the nanocomposite is extremely high, of the order of 1 V/µm and higher [59].

The value of *s_min_* is usually taken as 0.34 nm (van der Waals distance). It can be argued that adhesion between two nanotubes can decrease their minimal separation; for SWCNTs, the minimal distance caused by the adhesion was estimated as 0.25 nm [60].

The potential barrier Δ*E* is shown by [58] to have different values for the contact distances below and above the “polymer cutoff distance”, assumed to be 0.6 nm for polyethylene (PE) [58]. The same value, 0.6 nm, was used in [6,7,11,13] for epoxy as well and, in the absence of better estimations, is employed in the present calculations also. The difference between the potential barrier values below and above the polymer cutoff distance is attributed in [58] to a result of atomistic modelling. For a wider gap between the CNTs, the PE molecule penetrates into the gap, causing the decrease in the barrier height due to the resonance between tails of CNT wave functions and frontier orbitals in the PE molecule and leading to that for a charge that is easier (more probable) to transfer over the polymer-filled gap than over a vacuum gap. The values of Δ*E* are calculated in [58] for PE as Δ*E*_1_ (*s* < 0.6 nm) = 4.5 eV and Δ*E*_2_ (*s* > 0.6 nm) = 3.0 eV. Other authors provide values for Δ*E*_1_ down to 0.5–2.5 eV [61] and up to 5 eV [62].

#### 2.2.2. Number of Conductive Channels

Equation (2) is based on the approximation of an equivalent parallel-plate capacitor of area $d^{2}$ and separation *s*. An alternative formulation given by [63] and further used, for example, in [6,7,11,13], is:(4)$$G_{tunn}=G_{0}M\exp\left(-\tau s\right)$$
where *M* is the number of channels for tunneling. *M* = 2 for SWCNTs and *M* = 400–500 for MWCNTs [64]. Value *M* = 450 is generally employed in calculations [6,7,11,13]. Equalizing coefficients in Equations (2) and (4) at *s* = 0.34 nm and Δ*E* = 1–5 eV, the equivalent values of *M* for a SWCNT with *d* = 1.6 nm can be computed to be in the range 2–5, and for an MWCNT with *d* = 15 nm to be in the range 200–500. Therefore, Equation (2) provides the results close to the simplified Equation (4) but explicitly accounting for the dependency of coefficient *M* on the inter-CNT distance and CNT diameter. In the present calculations, Equation (2) will be used.

For numerical experiments in the present work, we will investigate the differences in homogenized conductivity and deformation sensitivity of nanocomposites, which originate from the difference in assumed Δ*E*_1_ values, in the interval 1–5 eV. In addition, it will be assumed that Δ*E*_2_ = Δ*E*_1_/1.5, and the polymer cutoff distance is 0.6 nm (Figure 1b).

## 3. Geometrical and Electrical Models

### 3.1. Geometrical Model

#### 3.1.1. Variants of CNT Geometry

Two geometrical arrangements of a wavy CNT assembly are considered here: (1) isotropically randomly oriented, uniformly spatially dispersed SWCNTs (designated “R-SW” below) and (2) aligned MWCNT assembly (A-MW), with parameters given in Table 1. Both types of nanocomposites were previously experimentally studied by the present authors [65,66,67,68,69].

#### 3.1.2. The Algorithm

The algorithm generating the geometrical model is described in [70,71] and is illustrated in Figure 2. It uses random choice of the direction of a generated CNT path segment, as is widely used in the literature, for example, in [72,73], but with certain constraints, as described below. Figure 2a shows a scheme of the *n*th segment of a CNT with length *l_seg_*, and orientation defined by the azimuthal angle *φ_n_* and polar angle *θ_n_*. To generate the aligned assembly, the angles *φ_n_* and *θ_n_* are defined in relation to the global Cartesian coordinate system of the assembly with axis *z* corresponding to the direction of forest growth. For generation of a random assembly, the angles are defined in the local coordinate system with axis *z* corresponding to the direction of the previous segment, where the first segment was oriented according to the uniform orientation distribution. The angles *φ_n_* and *θ_n_* are random values. They are first calculated as *φ_n_ =* rand(0,2π), cos*θ_n_ =* randN(1, cos(*σ_θ_*)), where rand(*a*,*b*) is a random generator of the uniform distribution on [*a*,*b*], randN(*a*, *σ*) is a random generator of the normal distribution with mean *a* and standard deviation *σ*, and *σ_θ_* is a characteristic polar angle deviation, which is calculated based on the length of the CNT segment and the maximal curvature. The randomness of segment orientation is restricted by the following conditions:
(1)Maximal path curvature and torsion are limited: $\kappa \leq \kappa_{max};\tau \leq \tau_{max}$. This type of control was introduced recently in [71] that demonstrated, apart from adequate representation of CNT shapes, that these conditions suppress the segment length dependence of the homogenized conductivity. If for generated *φ_n_* and *θ_n_* the curvature and torsion do not satisfy these conditions, *φ_n_* and *θ_n_* are generated again;(2)Correlated random angles: the sequences of *φ_n_* and *θ_n_* pairs are auto-correlated along the CNT path, with the assumed correlation length of 100 nm (see [70]).


For the random assembly, the origins of CNT paths are distributed uniformly in the model volume, and initial orientations are distributed isotropically. For the aligned assembly, CNTs’ origins are placed randomly and uniformly on a plane, using a Poisson random process.

For both cases, geometric periodicity is assumed; the model is confined within a representative volume element (RVE). If a CNT crosses an RVE face, then it is continued from the opposite face until the full length of the CNT is reached. The number of CNTs in the model is defined based on the prescribed volume fraction (VF). Figure 2a,b show random instances of (a) R-SW and (b) A-MW RVEs.

### 3.2. Electrical Model: Homogenized Conductivity

Once an RVE is created, the geometrical network of a CNT assembly is analyzed for contacts between CNTs and then transformed into a set of nodes, connected by electrical resistances/conductances, which are assigned to the tunneling contacts according to Equations (2)–(3) and to the CNT sections between the contacts according to Equation (1). The electrical boundary conditions are periodical. The homogenized conductivity tensor is then calculated using the homogenization theory outlined in Appendix A. The identification of contacts and nodal analysis are described in more detail in Appendix B.

### 3.3. Electrical Model: Piezoresistive Response

#### 3.3.1. Principles of the Model

At deformation, distances *s* of the tunneling contacts, Equation (2), are changing, causing the change in the tunneling part of conductivity. When modelled, care should be taken to apply correct values of potential barriers. As discussed in Section 2.2, in the undeformed state, the potential barrier Δ*E*_1_ is applied at distances shorter than the polymer cutoff distance 0.6 nm, while at larger distances the value of the potential barrier is switched to Δ*E*_2_. The reason behind this is that at distances between two CNTs larger than 0.6 nm, polymeric chains begin to penetrate into this gap, while at shorter distances it is prevented by van der Waals’ forces. When we apply deformation, the distances between CNTs change. However, since we consider only elastic polymers with no or little plasticity, we assume that no polymer interpenetrates the opening gaps, and the values of the potential barriers must be evaluated with the reference to the undeformed configuration. The following rules are followed:

(Rule 1): If at deformation the gap between two CNTs is opening and exceeds the polymer cutoff distance, the potential barrier still remains Δ*E*_1_.

On the contrary, when the gap is not opening, but closing, we expect van der Waals’ forces to push the polymer out of this gap, thereby changing the potential barrier. Hence:

(Rule 2): If at deformation the gap between two CNTs is closing and becomes shorter than the polymer cutoff distance, the potential barrier switches to Δ*E*_1_.

Finally, due to the lack of matrix plasticity, if CNTs are separated by a large distance, by default filled with polymer, no conformational changes, no matter how close moving one CNT to another, can form new contact points. Therefore, the third rule is:

(Rule 3): Pairs of contact points, at which tunneling is happening, stay unchanged at deformation, and in the deformed configuration, no search for new contact points is performed.

In the present algorithm, no contacts between CNTs are severed or moved along CNTs; only distances between CNTs change. In addition, no new contacts are created. Therefore, our modelling approach would be inapplicable to changes in the system of contacts in a dry filament assembly (e.g., buckypaper in case of CNT filaments or non-woven fibrous material without bonding). On the contrary, in our research, we are dealing with impregnated composite with quite limited applied strain range, where the presence of solid matrix prevents free creation of new contacts.

#### 3.3.2. Calculation of Gauge Factor

The piezoresistive gauge factor (*GF*) for uniaxial tension deformation *ε* in direction *i* is defined as
(5)$$GF=\frac{1}{\varepsilon }\frac{g_{ii}-g_{ii}^{\prime }\left(\varepsilon \right)}{g_{ii}}$$
where *g_ii_* and $g_{ii}^{\prime }\left(\varepsilon \right)$ are *ii* diagonal components of the homogenized conductivity tensors before and after the deformation.

The range of GF values found in the literature is very wide. Review [74] and later publications, e.g., [65,75], provide the following data. For non-aligned CNT distributions, with weight percentage of CNT below 2%, the most often observed values are in the range 1–7. Some authors give GFs up to 20–30 [76,77]. For aligned CNTs, the range of GFs for deformation in the direction of the CNT alignment is 0.04–10.

The simplest approach to model the deformational sensitivity is to consider dilatational deformational state in tension, assuming that all distances *s* of the tunneling contacts, Equation (2), change proportionally to the applied deformation (iso-strain) as *s’ = s*(1 + *ε*). In this case, the result can be estimated without the necessity of numerical simulations. Indeed, since all distances are increased proportionally and the values of potential berries are unchanged (rule 1), all tunneling conductances at contacts will drop proportionally to
(6)$$\frac{\exp\left(-\tau s\left(1+\varepsilon \right)\right)}{\exp\left(-\tau s\right)}\frac{s}{s\left(1+\varepsilon \right)}=\frac{\exp\left(-\tau s\varepsilon \right)}{1+\varepsilon }$$

Hypothesizing that the main contribution to the tunneling part of conductivity comes from distances shorter than the polymer cutoff distance (which will be proved later by Figure 3b), we can assume the value of τ in Equation (6) as being determined by Δ*E*_1_ = 4.5 eV, i.e., τ ~ 22 nm^−1^. For distances *s* between 0.34 and 0.6 nm and with *ε* = 0.01, we expect Equation (6) to be in the range from 0.87–0.93 with the GF of the tunneling part to be in the range from 7 to 13. This is the total GF in case of infinite intrinsic conductivity. For non-zero intrinsic resistivity, the GF will be lower. These values correspond to the experimental observations.

However, assuming a proportional increase in all tunneling distances, one assumes homogeneous deformation of a nanocomposite despite orders of magnitude difference in stiffness between CNTs and polymer. Therefore, for accurate analysis, conformational changes of CNTs must be considered. To avoid 3D finite-element simulations (see, e.g., [22]), we propose a simplified procedure, which calculates the change of the tunneling contact distances under the following assumptions:
Average deformation is transferred unchanged to sub-micrometer scale deformation elements around the contact points of CNTs;Length of a CNT does not change during the deformation.


The details of the applied deformation algorithm are given in Appendix C. The reader is referred to [78] for details of validation of the piezoresistivity model against finite elements calculations.

After new tunneling distances are calculated, the corresponding tunneling conductances are re-assigned.

## 4. Results and Discussion

### 4.1. Random RVE Instances

#### 4.1.1. RVE Sets and Orientation Distribution

Using the algorithm described in Section 3, the RVE sets are generated for R-SW and A-MW input data: 33 RVE instances of R-SW and 100 of A-MW. Examples of the RVEs are shown in Figure 2b,c. Potential differences are applied to all pairs on the opposite RVE faces in a sequence. After homogenized conductivity is calculated, this produces 99 independent values for principal conductivity tensor components for isotropic R-SW (33 RVE instances times 3 directions of the potential difference application), 100 and 200 independent values for *Z*-conductivity and X/Y-conductivity, respectively, for transversely isotropic A-MW, with *Z* being the direction of the CNT alignment.

The orientation tensors ***a*** for two particular RVE realizations, shown in Figure 2b,c, calculated based on averaging orientations of CNT segments, are
$$R\text{-}\mathrm{SW}:\;a=\left[\begin{array}{ccc} 0.346 & -0.005 & -0.014\\ -0.005 & 0.318 & -0.004\\ -0.014 & -0.004 & 0.336 \end{array}\right]$$
(7)$$A\text{-}\mathrm{MW}:\;a=\left[\begin{array}{ccc} 0.037 & -0.0003 & -0.013\\ -0.0003 & 0.037 & -0.0008\\ -0.013 & -0.0008 & 0.926 \end{array}\right]$$

The tensor for R-SW corresponds to the isotropic orientation, and for A-MW, the orientation is concentrated near the *z*-axis, with the average angle of deviation estimated as $\mathrm{acos}\left(\sqrt{a_{33}}\right)$ = 15.7°. These orientations are characteristic for all calculated RVE instances, with diagonal tensor components for R-SW deviating from 1/3 not more than by 0.02 and the average deviation angle for A-MW lying in the range 14–16°.

#### 4.1.2. Distribution of CNT Contacts

The conductivity of an RVE depends on contact distances between CNTs, which, in turn, define the tunneling resistances of contacts and the conductivity of CNT segments between the contacts. These characteristics are shown in Figure 3.

Figure 3a shows the number of CNT contacts *n_c_* per 1 µm^3^ in the random RVEs as a function of CNT volume fraction *VF* (an additional value *VF* = 2% is added to the calculated variants of R-SW for comparison). The simulated values are compared in Figure 3a with a theoretical formula from [35,36]:(8)$$n_{c}=\frac{16}{\pi^{2}}I\frac{VF^{2}}{d_{CNT}^{3}}k,\;\mathrm{where}\;k=\frac{d_{CNT}+0.0014\;µm}{d_{CNT}}$$

Here, *d_CNT_* is the CNT diameter, and *I* is an integral depending on the orientation distribution function; for isotropic distribution (R-SW), *I* = *π*/4, and for the distribution defined by the orientation tensor from Equation (7) for A-MW, *I* = 0.23, calculated using the formulae given in [35] for aligned wavy fibers. The correction coefficient *k* accounts for the fact that a contact is counted in the present simulations if the distance between CNT surfaces is in the range shorter than 1.4 nm, whilst in the theory, this contact distance is supposed to be zero. Comparison of the simulated values (dots in Figure 3a) with the theoretical trends shown by dashed lines, shows good correlation between the two. This allows us to conclude that the procedure for generation of CNT paths is consistent with general rules of geometry of random filament assemblies.

The scatter in the number of contacts for different random RVE realizations is minor (below ±10%) in all cases apart from the case of A-MW at low VF = 0.025; in the case of A-MW, the contact probability is defined by a combination of waviness-induced deviation of the CNT path from the straight vertical average line and a distance between these lines of adjacent CNTs, which is comparable to waviness deviations for low VF.

Each CNT contact is characterized by a contact distance between CNT surfaces, *s*. When the contacts are calculated, the distances *d_centr_* between the CNT centerlines (paths) are determined. A contact defines a pair of points on the centerlines such that distance *d_centr_* between them is smaller than the contact threshold of *d_CNT_* + 1.4 nm. The reader is referred to [71] for details of the contact calculations. Then, the distance between the CNT surfaces is calculated as
(9)$$s=\max\left(s_{min},d_{centr}-d_{CNT}\right)$$
where *s_min_* is the minimal van der Waals distance between two CNTs (see Section 2.2). The majority of the contacts have *d_centr_ < d_CNT_* + *s_min_*, hence *s = s_min_*. The relative number of contacts with *s > s_min_* increases with *s* because of the increase of cylindrical volume with radius (*d_CNT_ + s, d_CNT_ + s +* Δ*s*), where Δ*s* is the width of the histogram bin.

#### 4.1.3. Distribution of the Tunneling Conductances

Typical distributions of tunneling conductances of contacts are shown in Figure 3b for R-SW and A-MW cases. The overwhelming majority of contacts, 70–90%, are characterized by the maximal conductance, corresponding to the distance *s_min_*. The frequency of the histogram bins increases with the decrease in conductance value, corresponding to the increase of the relative number of contacts with higher *s*. The overall larger tunneling conductance in the MW case in comparison with SW corresponds to Equation (2) and curves shown in Figure 1b.

The tunneling resistance is compared with the mean resistance of CNT sections between the contacts (a distance between the contacts along a CNT). The distribution of lengths of these sections is shown in Figure 3c,d. Clearly, the mean values 60–90 nm for R-SW and 180–250 nm for A-MW are much larger than the characteristic defect-free length 4.7 nm [44]. Hence, CNT sections between two tunneling connections cannot be treated as defectless with the ballistic electron transport.

The mean inter-contact CNT resistance $R_{CNT}^{mean}$ is calculated inverting Equation (1) with *l_seg_ = b_mean_*, where *b_mean_* is the mean distance between two contacts along a CNT. The results of these calculations are plotted in Figure 4 (solid lines) as a dependency $R_{CNT}^{mean}\sim {\left(g_{intr}\right)}^{-1}$ for R-SW and A-MW cases, respectively. The levels for the tunneling resistance $R_{tunn}^{0.34}$, which is an inversion of the tunneling conductance given by Equations (2) and (3), are plotted on the same graphs for the range of Δ*E*_1_ = 1–5 eV and the minimal distance between the CNT surfaces of 0.34 nm (which corresponds to the overwhelming portion of the contacts, see Figure 3b). Depending on the *g_intr_* value, the relation between two resistances can be different. For *g_intr_* ~ 10^4^ S/m, $R_{CNT}^{mean}>R_{tunn}^{0.34}$, and the CNT, intrinsic resistance defines the homogenized resistivity/conductivity. On the other end of the studied range, for *g_intr_* ~ 10^8^ S/m, we have $R_{CNT}^{mean}\ll R_{tunn}^{0.34}$, and the tunneling resistance is the defining factor. In the middle of the *g_intr_* range, both resistances are comparable.

### 4.2. Plan of Numerical Experiments

To investigate the influence of the electrical conductivity input data on the homogenized RVE conductivity and deformation sensitivity, these parameters were calculated for random RVE instances, according to the plan of numerical experiments shown in Table 2. This plan investigates uncertainty of calculated performance characteristics of nanocomposites depending on (1) the uncertainty of setting the intrinsic conductivity of CNTs coupled with variation of the potential barrier for tunneling conductivity and (2) the uncertainty of the minimal (van der Waals) distance between CNTs (affected or not affected by CNT compression). The reference point of the plan is taken as
*g_intr_* = 10^4^ S/m; Δ*E*_1_ = 3 eV; *s_min_* = 0.34 nm
which is the set of parameters most used in the modelling publications.

### 4.3. Uncertainty in Homogenized Conductivity

#### 4.3.1. Conductivity at the Reference Point of the Plan for Numeric Experiments

Table 3 and Figure 5 and Figure 6 present the results of calculation of conductivity tensor for R-SW and A-MW configurations at the reference point of the plan for numerical experiments. Conductivity tensor $G=\left[G_{ij}\right]\;$(for its definition see Appendix A) is represented by the following group of values:

**R-SW**: This configuration is isotropic. Hence, all 3 diagonal components (*G*_11_*, G*_22_*, G*_33_) in all 33 random realizations of the RVE form a single sampling of 99 values; the same is valid for all 3 off-diagonal components (*G*_12_*, G*_13_*, G*_23_), which form a single sampling also with 99 values.

**A-MW**: This configuration is transversely isotropic. In all 100 random RVE realizations, conductivity along CNTs, *G*_33_, forms a sampling with size 100, conductivity across CNTs (*G*_11_*, G*_22_) forms a sampling with size 200, and for off-diagonal components (*G*_12_*, G*_13_*, G*_23_) a sampling with size 300 is formed.

The results of calculations present the expected dependency *G*(*VF*): with the CNT volume fraction above the percolation thresholds, both for R-SW and A-MW configurations, there is an increase of the homogenized conductivity with the increase of *VF*, but inside the same decimal order of magnitude of the conductivity values. This observation is typical post-percolation behavior well above the percolation threshold. The homogenized conductivity of A-MW in the CNT direction and across it differs by about 60 times.

The conductivity values, calculated using the reference point of the numerical plan (*g_intr_* = 10^4^ S/m; Δ*E*_1_ = 3 eV; *s_min_* = 0.34 nm), are within one order of magnitude difference with the experimentally measured values for the R-SW system [65] and the A-MW system [69]. The comparison with the experiment will be discussed more in detail after the variability of modelling with variation of the input is investigated.

The scatter of the modelled conductivity values, shown in Table 3 and Figure 5, is caused by random RVE geometry variations. For the diagonal components, the coefficient of variation is just a few percent. This stability of the homogenized conductivity for volume fractions well above the percolation thresholds for both systems corresponds to a stability of the number of contacts, which is seen in Figure 3a. The general tendency can be observed: the denser the percolation network, the lower the scatter.

#### 4.3.2. Dependency of the Homogenized Conductivity on the CNT Intrinsic Conductivity

Figure 7 shows results of simulations of the homogenized conductivity *G* (namely, different components of its tensor) for varying CNT intrinsic conductivity *g_intr_*. The other two variables of the numerical plan are kept constant at the reference values: Δ*E*_1_ = 3 eV, *s_min_* = 0.34 nm.

Figure 7 also shows a range of experimentally measured conductivity for R-SW and A-MW systems, reported by [65,69]. These ranges correspond to the results of the nodal analysis if the following values of *g_intr_* are assumed:R-SW: *g_intr_* = (1–2)⋅10^3^ S/m 
A-MW: *g_intr_* = (0.7–2)⋅10^4^ S/m 

Varying the intrinsic conductivity by one decimal order of magnitude leads to a change in the homogenized conductivity also by one order of magnitude in all modelled configurations (isotropic R-SW, A-MW along and across the bundle). For the intrinsic conductivity range 10^3^–10^6^ S/m, the dependency *G*(*g_intr_*) is linear in log–log coordinates of Figure 7 and is described by a power law
$$G\propto {\left(g_{intr}^{}\right)}^{p}$$
where power *p* equals 0.833 ± 0.005 for diagonal components of R-SW, and 0.982 ± 0.007 (along CNTs), 0.827 ± 0.005 (across CNTs) for A-MW (least square fit). It is noteworthy that all three values are below unity (the difference is much higher than the scatter range) which can be expected from 3D percolation below the upper critical dimension. If *g_intr_* reaches high values of 10^7^–10^8^ S/m, the growth of the dependency *G(g_intr_)* is slowed with the values of *G* bounded by a maximum corresponding to the infinite intrinsic conductivity of CNTs (ballistic electron transport), shown by dashed lines in Figure 7a–c.

The strong influence of *g_intr_* on the homogenized conductivity, evidenced by Figure 7, means that accurate choice of *g_intr_* plays a paramount role in the qualitative success of modelling.

As shown in Section 2.1 (see Figure 1a), the experimentally measured values of intrinsic conductivity differ up to two orders of magnitude in different experiments. Experiments with single CNTs reported conductivities around 10^6^ S/m for MWCNTs and near 10^8^ S/m for SWCNTs. Experiments with CNT bundles lead to lower values around 10^4^ S/m.

In numerical modelling (crosses in Figure 1a), in several cases, for example, [10,14,15], the value of 10^6^ S/m was assumed. At the same time, [10] stated that this value, based on CNT manufacturer’s datasheet (Nanocyl point in Figure 1a), leads to a gross overestimation of the calculated conductivity in comparison with experimental data. Authors in [14,15] gave a reference to [47] (see Figure 1a) as a foundation for the choice *g_intr_* = 10^6^ S/m for their modelling of carbon fibers/CNT hybrid composites. However, when comparing their calculations with experimental data on nanocomposites [79], with a good correspondence, [14,15] used a lower value *g_intr_* = 10^5^ S/m.

Extremely high values, in the order of *g_intr_* = 10^8^ S/m, measured by [46] for single SWCNTs (see Figure 1a), when put in the nodal analysis for an R-SW system (Figure 7a), led to the homogenized conductivity up to 10^3^–10^4^ S/m, VF = 0.5–1.0%. This is two to three orders of magnitude higher than values measured in [65,80,81,82].

Apparently, there is a contradiction between the results for intrinsic conductivity of CNTs obtained with experiments on single particles and with the correlations between experimentally measured conductivities of nanocomposites and their numerical counterparts. CNT aggregation and rope/bundle formation [83], absent in numerical studies, can cause the discrepancy between the predicted and experimentally measured conductivity of a nanocomposite, leading to the underestimation of the intrinsic conductivity. Therefore, we may hypothesize that in case of a detailed RVE modelling, with both agglomeration and bundling phenomena, values of the order of 10^6^ S/m for MWCNTs and 10^8^ S/m for SWCNTs may need to be implemented. However, implementation of these values in the modelling of as if perfectly dispersed and distributed particles (which corresponds to the present study) will certainly lead to the overestimation of the resulting conductivity.

Finally, for ordinary CNT production, the use of infinite conductivity looks like an oversimplification in spite of the considerable spread of this assumption [12,24,26,27,52,53,54,55]. Infinite CNT conductivity leads to the homogenized values several orders of magnitude higher in comparison with calculations implementing *g_intr_* = 10^4^ S/m. However, the justification of the infinite intrinsic CNT conductivity can be argued by special conditions of CNT manufacturing, providing defect-free structure.

#### 4.3.3. Dependency of the Homogenized Conductivity on the Tunneling Resistance Parameters and Minimal Inter-CNT Distance

Figure 8a–c show dependency of the homogenized conductivity *G*(Δ*E*_1_) on the potential barrier value Δ*E*_1_ of the tunneling conductance, Equations (2) and (3), for R-SW and A-MW cases at two levels of CNT volume fraction VF, and for three levels of the intrinsic CNT conductivity *g_intr_*, at *s_min_* = 0.34 nm. Behavior *G*(Δ*E*_1_) for different VFs is very similar: the homogenized conductivity decreases with the increase of Δ*E*_1_, the trend defined by the decrease of the tunneling conductance with increase of Δ*E*_1_, see Figure 1b.

The slope of *G*(Δ*E*_1_) depends on *g_intr_*. For *g_intr_* = 10^4^ S/m, there is almost no dependency of *G* on Δ*E*_1_: it equals ~10% change in the range Δ*E*_1_ = 1–3 eV. The CNT segment resistance is higher than the tunneling resistance in this case (see Figure 3c,d), and the latter (defined by Δ*E*_1_) only weakly influences the homogenized conductivity.

For *g_intr_* = 10^6^ S/m, the dependency *G*(Δ*E*_1_) becomes more pronounced. In the range Δ*E*_1_ = 1–3 eV, the homogenized conductivity changes by about five times for R-SW and A-MW across CNTs; in these two cases, the number of contacts on a percolating path is considerable. For *g_intr_* = 10^6^ S/m, the CNT segment resistance and the tunneling resistance are in the same range and affect the homogenized conductivity collectively. For A-MW along CNTs, the change of *G*(Δ*E*_1_) over Δ*E*_1_ = 1–3 eV is minor, about 20%. This is explained by the fact that in the longitudinal direction of the bundle, there are only 2–3 contacts on the percolating path between two RVE sides, and the conductivity is defined by *g_intr_*, not tunneling.

The maximal slope of *G*(Δ*E*_1_) is reached at extremely high *g_intr_* = 10^8^ S/m, which is almost equivalent to the infinite CNT conductivity assumption (see Figure 7). For this case, the CNT segment resistance is much lower than the tunneling resistance, Figure 4. The resistance of a percolating path is defined by resistances of tunneling contacts included in the path and hence strongly depends on Δ*E*_1_*. G*(Δ*E*_1_) changes over Δ*E*_1_ = 1–3 eV by 10 (A-MW) to 50 (R-SW) times.

Figure 8d–f show dependency of the homogenized conductivity *G*(Δ*E*_1_) on the minimal inter-CNT distance *s_min_*, for R-SW and A-MW cases at two levels of the CNT volume fraction VF and for three levels of the intrinsic CNT conductivity *g_intr_*, at Δ*E*_1_ = 3 eV; *s_min_* has two levels: *s_min_* = 0.34 nm, which is van der Waals distance, and 0.25 nm, given by [60] as a result of CNT lateral compression. Change in *s_min_* is equivalent to change of $\sqrt{\Delta E_{1}}$ (see Equations (2) and (3)); hence, there is no surprise that the curves in Figure 8d–f show the behavior very similar to Figure 8a–c. Varying the minimal inter-CNT distance is felt only in the case of very high *g_intr_*.

A practical conclusion can be drawn from our modeling observations. If non-infinite intrinsic CNT conductivity is assumed, with a value from the range 10^4^–10^6^ S/m, then the choice of Δ*E*_1_ value (in the range 1–5 eV) does not significantly affect the homogenized conductivity. In contrast, if there are reasons to assume infinite (very high) intrinsic CNT conductivity (defectless CNTs), then the choice of Δ*E*_1_ affects the calculated homogenized conductivity strongly. The similar conclusions were reached in studies of conductivity of CNT films [84] and for random CNT assemblies with Vf < 2% in [32], but further experimental observations are desired since it contradicts experimental evidence and theoretical arguments on the generally prevailing role of the tunneling conductivity [12,85].

The relative influence of the intrinsic conductivity and tunneling barrier parameters plays an important role when temperature dependence of nanocomposites conductivity is being discussed. Both random and aligned nanocomposites exhibit negative temperature coefficients of resistance (TCR), with resistivity decreasing by 5–15% when temperature increases from room temperature to 400–500 K [84,85,86,87,88]; the same behavior was noted in nanoplatelet networks [89,90]. This decrease in resistivity is governed by temperature dependence of both factors defining the homogenized conductivity: the tunneling resistance [85,90,91] and the intrinsic conductivity [34,91], combined with the thermal expansion. Further experimental evidence and a model of such processes are yet to be discovered.

### 4.4. Uncertainty of Deformation Sensitivity

#### 4.4.1. Deformation Sensitivity at the Reference Point of the Numerical Plan

Table 4 and Figure 9 present the results of calculation of the gauge factors GF for R-SW and A-MW configurations at the reference point of the plan for numerical experiments, with uniaxial tension strain *ε* = 0.01. For A-MW, two cases are considered: deformation along and across aligned CNTs. In all cases, the direction of applied voltage coincides with the direction of the applied deformation (piezoresistivity response is calculated in the direction of the deformation).

The calculated gauge factor values with parameters at the reference point are within the range of the experimentally observed GFs [65,74,75], as illustrated in Figure 9d. In contrast to dilatational loading (6), dependencies GF(VF) appear. The calculated trend GF(VF) is decreasing both for R-SW and A-MW (deformation along CNTs), which also corresponds to the literature data [65,76].

For A-MW deformation across CNTs, we observe anomalous behavior when the increase in VF does not lead to significant change in GF (the distributions of GFs overlap for VFs 2.5% and 7%). Calculations show that in this case, contrary to R-SW and A-MW deformed along CNTs, the relative change of the tunneling distances *ds = (s’ − s)/s* is distributed in a narrow range, *ds* < 1 (*s’* is the tunneling distance after the deformation, *s* is the distance in the undeformed configuration). This is not far from a dilatational deformation, described by Equation (6), for which the change of conductivity is independent of VF. For all other cases (R-SW and A-MW along CNTs), the distributions of *ds* are much wider, with *ds* up to 100.

The calculated GF values show low scatter, with the coefficient of variation of a few percent, similar to the homogenized conductivity values (Table 3 and Figure 5). An exception is A-MW, Vf = 2.5% case, where the GF distribution for both cases along/across CNTs is wide. It is probably a result of a comparatively low number of inter-CNT contacts in this case (see Figure 3a); hence, there is a possibility for large variations of a percolating network. However, the general tendency is clear: the denser the percolation network is, and, thereby, the farther we are from the percolation threshold as a point of a phase transition, the less fluctuating behavior we observe. This tendency is related to systems with phase transitions: the closer the critical point (percolation threshold in this case) is, the more fluctuating behavior of a system becomes [92]. The decrease of the coefficient of variation for the gauge factor with the increase of the CNT volume fraction was also predicted in [33] for the aligned CNT case.

#### 4.4.2. Dependency of Gauge Factors on Intrinsic CNT Conductivity and Tunneling Resistance

Figure 10 and Figure 11 show dependency of the gauge factors GF on the assumed parameters: the intrinsic conductivity of CNTs and potential barrier Δ*E*_1_ for the tunneling conductivity. GF grows both with the increase of *g_intr_* and with the increase of Δ*E*_1_. The dependency on the intrinsic CNT conductivity (Figure 10) can be explained by the fact that with the increase in *g_intr_* the homogenized resistivity is more strongly determined by the tunneling resistances, sensitive to deformation and, thereby, providing higher GFs. The second dependency, on the value of the potential barrier (Figure 11), is explained by the fact that tunneling resistances, determining the piezoresistive GF, increase exponentially with the increase in the square root of Δ*E*_1_. The higher Δ*E*_1_, the steeper the slope of this dependence is, providing higher GFs. In contrast to the dependency *G*(*g_intr_*,Δ*E*_1_), at low *g_intr_* = 10^4^ S/m the change of Δ*E*_1_ in the range 1–5 eV changes GF by a factor 2–5 both for R-SW and A-MW, along and across CNTs. For *G*(10^4^
*S/m*, Δ*E*_1_), this factor was 1.1–1.2 (Figure 8).

It is interesting to note that GF values for R-SW and A-MW across CNTs are quite close one to another (within ~25% difference), even in spite of the difference in VF.

Influence of the uncertainty of the input parameters on GF, calculated with the nodal analysis of CNT networks, can be characterized as follows:
-For the same Δ*E*_1_, a change of GF is approximately proportional to a change of log(*g_intr_*);-For the same *g_intr_*, a change of GF is approximately proportional to a change of Δ*E*_1_.


## 5. Conclusions

We have conducted numerical experiments to quantify the uncertainty of the nodal analysis results for modelling of homogenized electrical conductivity tensor and deformation gauge factor for two types of CNT-based nanocomposites: randomly dispersed SWCNTs and aligned MWCNTs in a polymer matrix. The uncertainty is caused by the uncertain choice of the input parameters for analysis: the intrinsic conductivity of CNTs *g_intr_* and potential barrier Δ*E*_1_ for the tunneling resistance. The volume fraction (VF) of CNTs was chosen above the percolation threshold for all cases (0.5% and 1% for R-SW; 2.5% and 7% for A-MW).

The results of numerical experiments can be summarized as follows:
The scatter of the homogenized conductivity and gauge factor, caused by the randomness of a CNT assembly, is limited to few percent of coefficient of variation (CV) and is much smaller that the variations which may be caused by the difference in assumed values of the input parameters.The influence of the assumed intrinsic conductivity on the homogenized conductivity and gauge factor is strong. For the intrinsic conductivity range 10^3^–10^6^ S/m, the dependency *G*(*g_intr_*) is nearly linear (to the power-law dependence with the exponent being close but below unity). For higher *g_intr_* values, its growth is asymptotically limited by the conductivity corresponding to the case of infinite intrinsic conductivity of CNTs. The assumption of infinite CNT intrinsic conductivity may bring unrealistically high simulated values for the homogenized conductivity.The choice of the CNT intrinsic conductivity ~10^4^ S/m brings the simulated homogenized conductivity and gauge factor within one order of magnitude closeness to the experimentally measured values; a better choice asks for tuning of this parameter.The influence of the assumed potential barrier Δ*E*_1_ of the tunneling resistance on the homogenized conductivity is relatively weak for *g_intr_* ~ 10^4^–10^6^ S/m (limited to tens of percent variation for Δ*E*_1_ in the range 1–5 eV). The influence of Δ*E*_1_ becomes stronger if there are reasons to assume higher CNT intrinsic conductivity values.The influence of both *g_intr_* and Δ*E*_1_ on the simulated gauge factor is strong, about one order of magnitude span for the studied range of the input parameters.


Strong dependencies of the homogenization results on the input parameters stress the necessity to evaluate a possible range of particular simulation results on the parameter choice. The findings may affect methods for optimization of nanocomposites morphology, highlighting the need to use fuzzy target functions.

## Figures and Tables

**Figure 1 polymers-14-04794-f001:**
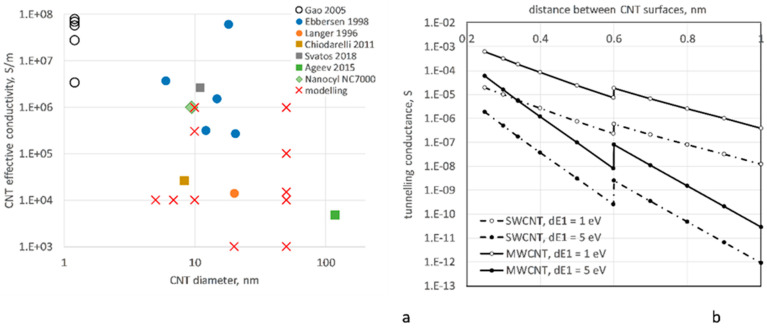
Uncertainty in electrical conductivity input data: (**a**) experimental data on intrinsic CNT conductivity, SWCNT (open symbols [46]), and MWCNT (filled symbols [44,47,48,49,50,51]); round points: experiments with single CNTs; squares: experiments with aligned CNTs; diamond: manufacturer’s datasheet; crosses: input data used in various modelling works; (**b**) tunneling conductance of a contact between two CNTs in function of the distance between them; results for SWCNTs and MWCNTs are calculated with Equations (2)–(4).

**Figure 2 polymers-14-04794-f002:**
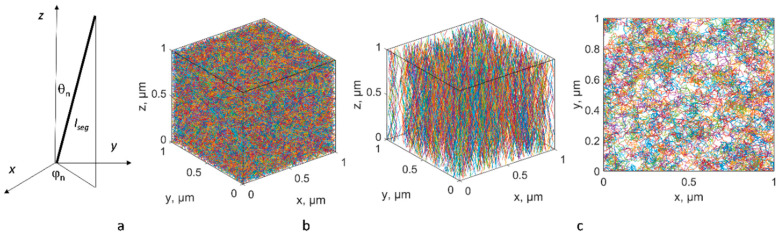
Geometrical model: (**a**) parameters of a CNT segment; central paths of CNTs in (**b**) RVE for R-SW, VF = 1%; (**c**) RVE for A-MW; VF = 7%, 3D view (**left**) and view from the end of the *Z*-axis (**right**).

**Figure 3 polymers-14-04794-f003:**
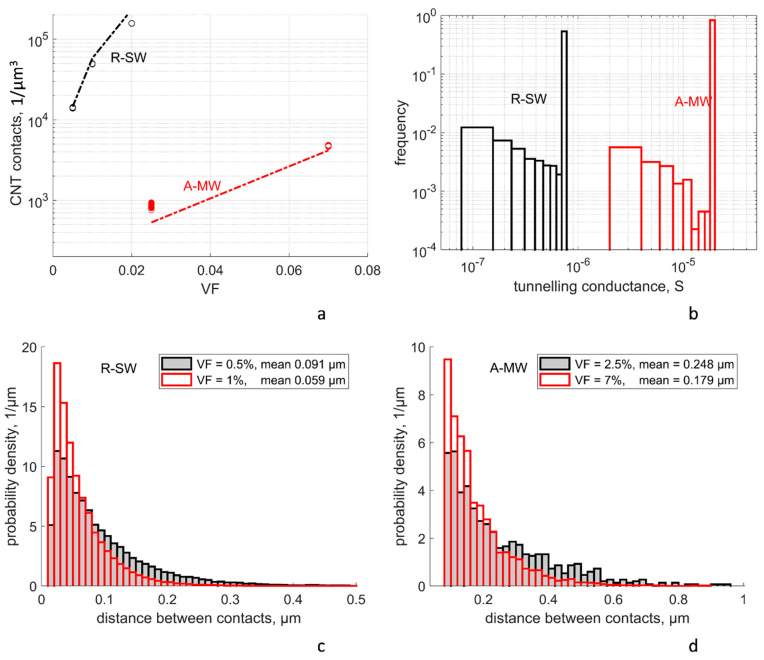
Characteristics of contacts for random RVEs: (**a**) density of CNT contacts, points: calculated values for random RVEs (the scatter is not noticeable for R-SW), lines: prediction according to Equation (8); (**b**) distribution of the tunneling conductances of CNT contacts, Δ*E*_1_ = 3 eV, *s_min_* = 0.34 nm, typical RVEs for R-SW (VF = 1%) and A-MW (VF = 7%) cases; (**c**,**d**) distribution of distances along CNTs between the contacts for R-SW (**c**) and A-MW (**d**) cases; mean values are shown in the legends.

**Figure 4 polymers-14-04794-f004:**
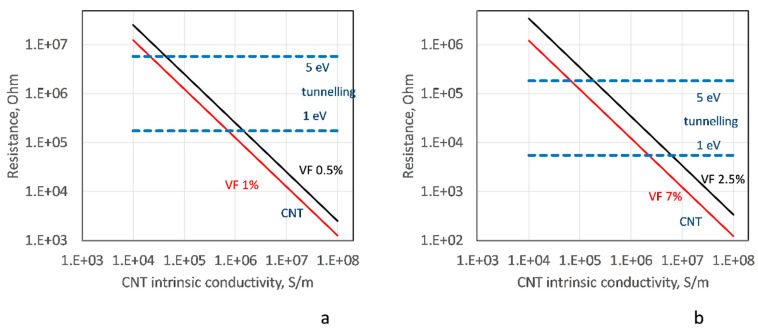
Comparison between the resistance of a typical CNT segment between two contacts for two values of VF (solid lines) and the tunneling resistance in the contact for s = 0.34 nm for two values of Δ*E*_1_ (dashed lines), R-SW (**a**) and A-MW (**b**).

**Figure 5 polymers-14-04794-f005:**
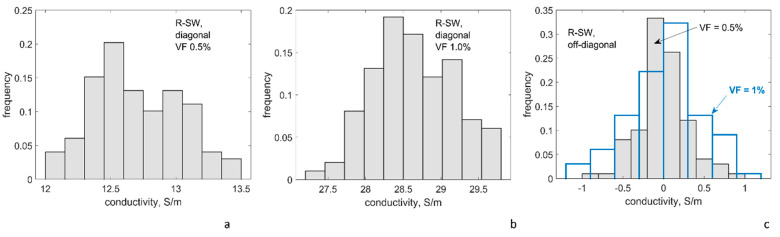
Conductivity tensor components for R-SW, histograms of the distributions: diagonal VF = 0.5% (**a**); VF = 1% (**b**); and off-diagonal VF = 0.5% and 1% components (**c**); the reference point of the numerical plan: *g_intr_* = 10^4^ S/m; Δ*E*_1_ = 3 eV; *s_min_* = 0.34 nm; sampling size ≥ 99 for all cases.

**Figure 6 polymers-14-04794-f006:**
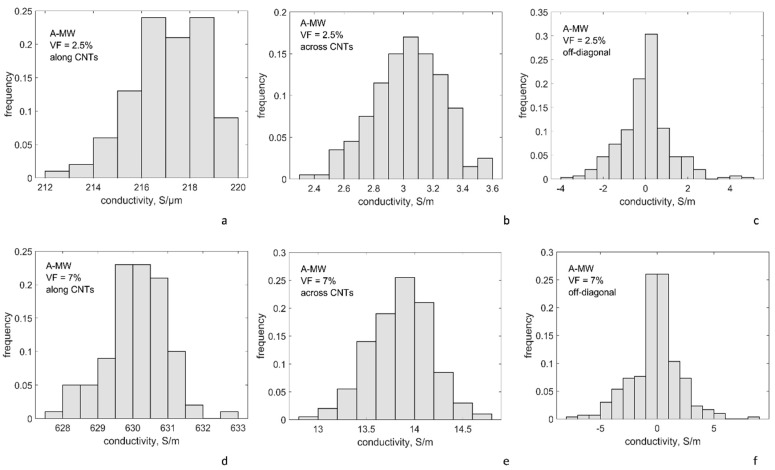
Conductivity tensor components for A-MW, histograms of the distributions: VF = 2.5% (**a**–**c**); and VF = 7% (**d**–**f**); conductivity along CNTs (**a**,**d**); across CNTs (**b**,**e**); and off-diagonal (**c**,**f**); the reference point of the numerical plan: *g_intr_* = 10^4^ S/m; Δ*E*_1_ = 3 eV; *s_min_* = 0.34 nm; sampling size ≥ 99 for all cases.

**Figure 7 polymers-14-04794-f007:**
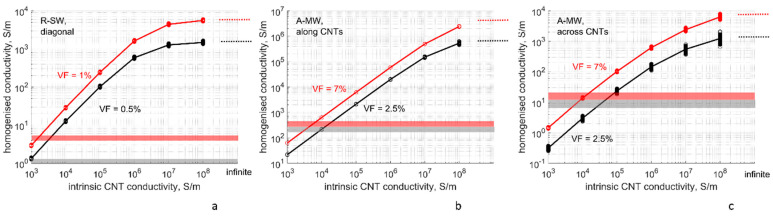
The homogenized conductivity as a function of the assumed intrinsic conductivity of CNTs, at two levels of the CNT volume fraction: (**a**) R-SW, diagonal components of the conductivity tensor; (**b**) A-MW, along CNTs; (**c**) A-MW, across CNTs. The dashed lines show homogenized conductivity calculated assuming infinite conductivity of CNTs. The points show conductivity values, corresponding to different RVE random realizations (sampling size ≥ 99 for all cases). Δ*E*_1_ = 3 eV, *s_min_* = 0.34 nm, the lines show mean values. The horizontal boxes show the homogenized conductivity range measured for R-SW [65] and A-MW [69] experimentally. The colors of the lines, points, and boxes correspond to CNT volume fractions, as shown on the graphs.

**Figure 8 polymers-14-04794-f008:**
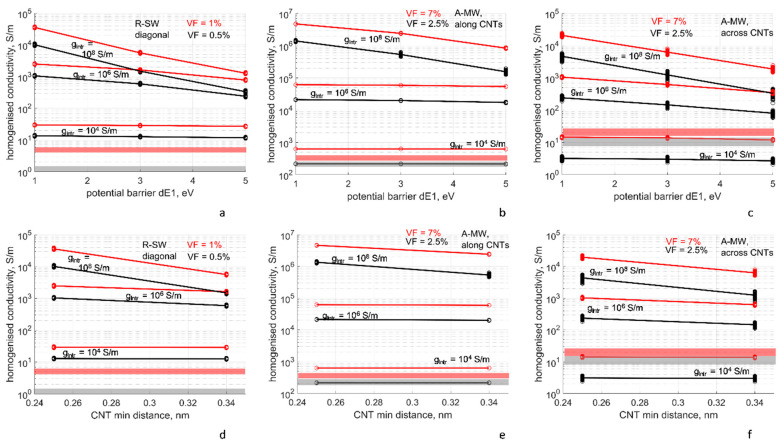
The homogenized conductivity as function (**a**–**c**) of the assumed potential barrier Δ*E*_1_ for the tunneling conductivity, with *s_min_* = 0.34 nm, and (**d**–**f**) of the minimal distance between CNTs, with Δ*E*_1_ = 3 eV, at three levels of the CNT intrinsic conductivity *g_intr_* and two levels of the CNT volume fraction: (**a**,**d**) R-SW, diagonal components of the conductivity tensor; (**b**,**e**) A-MW, along CNTs; (**c**,**f**) A-MW, across CNTs. The points show conductivity values corresponding to different RVE random realizations (sampling size ≥ 99 for all cases); the lines show mean values. The horizontal boxes show the homogenized conductivity range measured for R-SW [65] and A-MW [69] experimentally. The colors of the lines, points, and boxes correspond to CNT volume fractions, as shown on the graphs.

**Figure 9 polymers-14-04794-f009:**
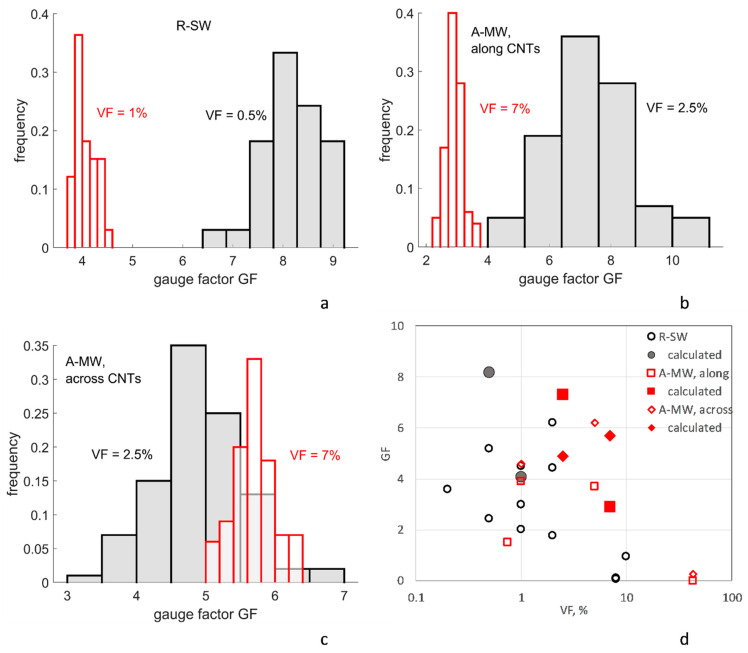
The gauge factors, Equation (5), *ε* = 0.01, histograms of the distributions: (**a**) R-SW, VF = 0.5% and 1%; (**b**,**c**) A-MW, along (**b**) and across (**c**) CNTs, VF = 2.5% and 7%; (**d**) experimental data [65,74,75], open symbols, in comparison with the computed values (closed symbols). The reference point of the experimental plan: *g_intr_* = 10^4^ S/m; Δ*E*_1_ = 3 eV; *s_min_* = 0.34 nm. Sampling size ≥ 99 for all cases.

**Figure 10 polymers-14-04794-f010:**
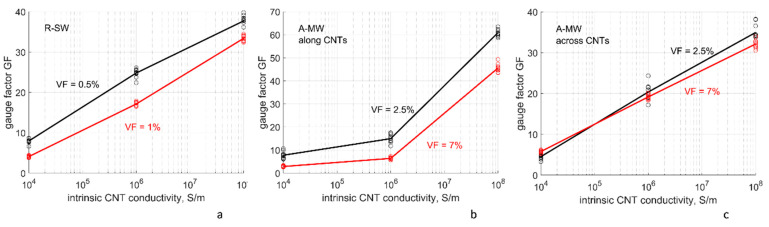
The gauge factors as a function of the assumed intrinsic conductivity of the CNTs, at two levels of the CNT volume fraction: (**a**) R-SW; (**b**) A-MW, along CNTs; (**c**) A-MW across CNTs. The points show conductivity values corresponding to different RVE random realizations; the lines show mean values; Δ*E*_1_ = 3 eV; *s_min_* = 0.34 nm.

**Figure 11 polymers-14-04794-f011:**
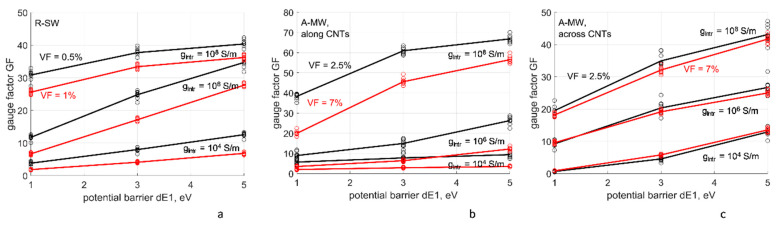
The gauge factors as functions of the assumed potential barrier Δ*E*_1_ for the tunneling conductivity, *s_min_* = 0.34 nm, at three levels of the CNT intrinsic conductivity *g_intr_* and two levels of the CNT volume fraction: (**a**) R-SW; (**b**,**c**) A-MW: along CNTs (**b**) and across CNTs (**c**). The points show conductivity values, corresponding to different RVE random realizations; the lines show mean values.

**Table 1 polymers-14-04794-t001:** Parameters of the modelled CNT nanocomposites.

Parameter	R-SW	A-MW
Data source	[65]	[66,68]
CNT outer diameter, nm	1.6	8.0
CNT length		
Distribution type	Weibull	constant
Mean length, µm	2.0	20
Weibull modulus	3.0	n/a
Weibull scale, µm	2.24	n/a
CNT orientation		
Distribution type	uniform	aligned
CNT shape		
Maximal curvature, 1/µm	5	5
Maximal torsion, 1/µm	5	5
Volume fraction variants	0.5%; 1%	2.5%; 7%

**Table 2 polymers-14-04794-t002:** Plan of numerical experiments, values in **bold** correspond to the reference point.

Parameter Variation Type	Intrinsic CNT Conductivity, S/M	Potential Barrier Δ*E*_1_, eV	Minimal CNT Distance, Nm
Varying intrinsic CNT conductivity and potential barrier for tunneling conductivity	10^3^ **10^4^** 10^5^ 10^6^ 10^7^ 10^8^ ∞	1 **3** 5	**0.34**
Varying minimal CNT distance	**10^4^**	**3**	0.25

**Table 3 polymers-14-04794-t003:** Homogenized conductivity tensor components at the reference point of the numerical plan: *g_intr_* = 10^4^ S/m, Δ*E*_1_ = 3 eV, and *s_min_* = 0.34 nm.

Type	VF	Diagonal Components, S/M		Off-Diagonal Components, S/M	
				Signed Values	Absolute Values
R-SW	0.5%	12.7 ± 0.33 (2.5%)		−0.007 ± 0.290	0.217 ± 0.195 (87%)
	1%	28.6 ± 0.55 (1.9%)		0.030 ± 0.422	0.334 ± 0.258 (77%)
A-MW		along CNTs	across CNTs		
	2.5%	217 ± 1.44 (0.66%)	3.03 ± 0.23 (7.5%)	−0.0015 ± 1.20	0.816 ± 0.874 (107%)
	7.0%	630 ± 0.86 (0.13%)	13.8 ± 0.32 (2.3%)	−0.140 ± 2.11	1.47± 1.52 (103%)

Notes: ± means standard deviation; the value in brackets gives coefficient of variation (not given for signed values of the non-diagonal components as their mean is close to zero).

**Table 4 polymers-14-04794-t004:** Gauge factors, Equation (5), for unidirectional tension deformation 0.01 in Z-direction, the reference point of the numerical plan: *g_intr_* = 10^4^ S/m, Δ*E*_1_ = 3 eV, *s_min_* = 0.34 nm.

Type	VF	Gauge Factor GF
R-SW	0.5%	8.18 ± 0.57 (7.0%)
	1%	4.07 ± 0.21 (2.5%)
A-MW, along CNTs	2.5%	7.32 ± 1.34 (18.3%)
	7.0%	2.91 ± 0.27 (9.3%)
A-MW, across CNTs	2.5%	4.89 ± 0.62 (12.6%)
	7.0%	5.69 ± 0.28 (4.9%)

*Note:* ± means standard deviation; the value in brackets gives coefficient of variation.

## Data Availability

The raw/processed data required to reproduce these findings cannot be shared at this time as the data also forms part of an ongoing study.

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
