# Peer review of "Uncertainties in Electric Circuit Analysis of Anisotropic Electrical Conductivity and Piezoresistivity of Carbon Nanotube Nanocomposites"

_polymers, 2022, doi:10.3390/polym14224794_

Round 1

Reviewer 1 Report

This is an interesting study that evaluates the reliability of nodal analysis on the electric conductivity of CNT composites with different features. Real-world parameters (including the intrinsic CNT conductivity, nanocomposite randomness, and degree of defects in CNT) and their influence on the modelling uncertainties are considered, which is very important to the understanding and evaluation of models for practical use. Such empirical verification can provide valuable guidance for future model improvements. Meanwhile, the manuscript is well-written, and the conclusions are logically proven with results. Publication is recommended.

Author Response

polymers-1955902          Lomov S.V., Gudkov N.A., Abaimov S.G

ANSWERS TO REVIEWERS

Reviewer 1

Reviewer’s comment

Author’s answer

This is an interesting study that evaluates the reliability of nodal analysis on the electric conductivity of CNT composites with different features. Real-world parameters (including the intrinsic CNT conductivity, nanocomposite randomness, and degree of defects in CNT) and their influence on the modelling uncertainties are considered, which is very important to the understanding and evaluation of models for practical use. Such empirical verification can provide valuable guidance for future model improvements. Meanwhile, the manuscript is well-written, and the conclusions are logically proven with results. Publication is recommended.

We appreciate the positive evaluation of our work.

Extensive English polishing has been conducted.

Reviewer 2 Report

Herein, the authors assess the electrical conductivity of carbon nanotubes with respect to various factors such as their diameter, distance, and distribution. The exposition is clean and explanations of the resulting data thorough; however, it is my opinion that the article may be rather suitable for another MDPI journal such as Materials.

I have minor amendments below:

Title: Change: nanotubes to nanotube

Line 57: Change: place period punctuation after the citation bracket.

Line 85: Change: [refs]; list the references

Author Response

polymers-1955902          Lomov S.V., Gudkov N.A., Abaimov S.G

ANSWERS TO REVIEWERS

Reviewer 2

Reviewer’s comment

Author’s answer

Herein, the authors assess the electrical conductivity of carbon nanotubes with respect to various factors such as their diameter, distance, and distribution. The exposition is clean and explanations of the resulting data thorough; however, it is my opinion that the article may be rather suitable for another MDPI journal such as Materials.

We appreciate the positive assessment of our work.

We think that the paper fits in the scope of Polymers in general, namely “Polymer Physics and Theory” and “Polymer composites and Nanocomposites”, and in the special issue on “Polymers’ Role in Sensors”.

We implemented and answered all corrections (for convenience, highlighted in green). Extensive English polishing has been conducted.

Title: Change: nanotubes to nanotube

corrected

Line 57: Change: place period punctuation after the citation bracket.

We have corrected the sentence structure adding word “study” before the reference.

Line 85: Change: [refs]; list the references

references added

Reviewer 3 Report

The paper presents many details and there is a lot of discussion regarding the methodology and the results. Being so detail intensive, the reviewer recommends putting a little more effort into improved presentation for the reader since it is hard to keep track of the many parameters and variations in these. Suggestions include subdividing sections further and breaking up some long sentences to improve readability.

1.     51 The reviewer agrees with the authors that study of uncertainty in input parameters is a subject that has not been touched upon extensively while studying the electrical properties of CNT/polymer nanocomposites. However, there have been some studies conducted on capturing the stochastic variation of effective electrical properties. For example, see “Statistical Analysis … “ Talamadupula and Seidel 2021, Computational Materials Science, where the distribution in effective properties was studied with respect to varying barrier potential and CNT dispersion.

2.     85 [refs]?

3.     74 Consider splitting up this sentence as the meaning is not very clear.

4.     88 Is this a supposition made by the authors or has it already been studied by others?

5.     75 to 79: The authors should elaborate on why the typical distance between CNTs being around 0.5 microns means that the conductivity along CNT segments between two contact points should be taken as ballistic. This is not clear.

6.     Figure 1b: Maybe this will be touched upon later but the immediate question here is why is there a jump in the tunneling conductance at 0.6 nm?

7.     118 This correlation is not readily apparent from the plot. Perhaps a least squares fit to confirm this would be appropriate.

8.     119 It would be interesting to see if there has been any other works that have suggested as such.

9.     161 This probably explains the jump in figure 1b. That figure may be better suited in this section instead of the previous one.

10.  Equation 4: The text here is a bit confusing. Is the present model going to be using equation 2 or equation 4? Is equation 4 presented just for comparison purposes to show the dependency of a supposed M coefficient?

11. Equation 4: The inference of the potential barrier  is slightly confusing. Clearly, from equation 2 and 3, and also from figure 1b, with larger values of barrier potential, the tunneling conductance reduces for the same distance. But then, how is it that  has a lower value than ? Is  supposed to be interpreted as the summation of these sequential barrier potentials as distance increases?

12.  159 This sentence is concerning. If polymer molecules were able to enter the gap, would that not increase the tunneling barrier? Or is there some other physics going on here that acts to reduce the tunneling barrier?

13. Figure 1b: If  is different with d < 0.6 and d > 0.6, wouldn’t the line slopes be appreciably different before and after d=0.6nm for all cases?

14.  176 Is this relationship between  and  typical? What about the selection of polymer cutoff distance as 0.6nm? Is there any further justification for these choices?

15.  223 This is a slightly more detailed explanation than when the concept was introduced in the earlier section. Should have the more detailed explanation in the earlier section.

16.  250 What is the original threshold for the maximum distance at which contact points are identified? If there is one, would there not be new contact pairs in the deformed configuration? The reverse may also be true, with pairs exiting out of range for conduction. Are there scenarios accounted for in the model, and if not, can the authors discuss the potential consequence of including/excluding this?

17.  759 Is this an assumption being made? If so, can the authors comment further on this and its limitations?

18.  Where is figure Cd?

19.  Table 1: Has this table been discussed anywhere in the manuscript?

20.  322 Where exactly is this correlation being made? It is not clear.

21.  334 The calculation of s is not clear. What do the authors mean by distance between the CNT centerlines s_center? Why is the contact distance taken as the maximum of either s_min or s_centr – d_CNT? Is this approach the reason why the histogram shown in figure 3b shows a large frequency bin towards the higher conductance side for both R-SW and A-MW?

22.  344 This line is not clear. What is a “section of a CNT from one tunneling connection to the next, located farther along this CNT”? Does it plainly mean the distance between contacts?

23.  350 Is some part of the sentence missing here?

24.  Figure 4: Is the caption correct? Where are e and d?

25.  Figure 6: Is the caption correct? Where is g, h , i?

26.  443 plays a paramount role

27.  584 increasingly instead of more and more

28.  The conclusion in general is a bit too detailed. Please work on summarizing.

29.  630 This explanation should’ve been there in the results discussion, if it is not there already. The discussion about scatter also may not be necessary in the conclusion.

Author Response

polymers-1955902          Lomov S.V., Gudkov N.A., Abaimov S.G

ANSWERS TO REVIEWERS

Reviewer 3

Reviewer’s comment

Author’s answer

The paper presents many details and there is a lot of discussion regarding the methodology and the results. Being so detail intensive, the reviewer recommends putting a little more effort into improved presentation for the reader since it is hard to keep track of the many parameters and variations in these. Suggestions include subdividing sections further and breaking up some long sentences to improve readability.

We have added the third level of headings systematically in all sections. Long sentences are systematically shortened.

All corrections are implemented and answered (for convenience, they are highlighted in blue)

Extensive English polishing has been conducted.

1. 51 The reviewer agrees with the authors that study of uncertainty in input parameters is a subject that has not been touched upon extensively while studying the electrical properties of CNT/polymer nanocomposites. However, there have been some studies conducted on capturing the stochastic variation of effective electrical properties. For example, see “Statistical Analysis … “ Talamadupula and Seidel 2021, Computational Materials Science, where the distribution in effective properties was studied with respect to varying barrier potential and CNT dispersion.

Thank you for the recent reference, which is very relevant to our work. It is added in the discussion in Introduction and in discussion of the results (scatter of the gauge factors).

2. 85 [refs]?

references added

3. 74 Consider splitting up this sentence as the meaning is not very clear.

corrected

4. 88 Is this a supposition made by the authors or has it already been studied by others?

the reference to [44] is repeated in this sentence for clarity.

5. 75 to 79: The authors should elaborate on why the typical distance between CNTs being around 0.5 microns means that the conductivity along CNT segments between two contact points should be taken as ballistic. This is not clear.

This is not a distance between CNTs, but distance between contact points along a CNT, which is estimated to be shorter than the free path. A clarification, also for definition of b, is added.

6. Figure 1b: Maybe this will be touched upon later but the immediate question here is why is there a jump in the tunneling conductance at 0.6 nm?

This is explained in section 2.2.1 and related to the change in the potential barrier due to the ability of polymer penetrate in-between the CNTs. We think it is useful to have two parts of Fig 1 together, even if Fig 1b is discussed later.

7. 118 This correlation is not readily apparent from the plot. Perhaps a least squares fit to confirm this would be appropriate.

As it is said, the correlation (downwards trend) is weak. Adding a linear regression to the plot would give too much significance to it in our opinion.

8. 119 It would be interesting to see if there has been any other works that have suggested as such.

We have not seen such statement (big diameter – more defects – low gintr) in literature, so this is our speculation. However, there are statements in the literature (for example [44]) that gintr can be used as a measure for the growth quality, which points to this direction.

9. 161 This probably explains the jump in figure 1b. That figure may be better suited in this section instead of the previous one.

Yes, this is correct. As said in an answer to question 6, we think it is useful to have two parts of Fig 1 together, even if Fig 1b is discussed later here.

10. Equation 4: The text here is a bit confusing. Is the present model going to be using equation 2 or equation 4? Is equation 4 presented just for comparison purposes to show the dependency of a supposed M coefficient?

Indeed, eq (2) is used. Clarification is added.

11. Equation 4: The inference of the potential barrier Δ? is slightly confusing. Clearly, from equation 2 and 3, and also from figure 1b, with larger values of barrier potential, the tunneling conductance reduces for the same distance. But then, how is it that Δ?2 has a lower value than Δ?1? Is Δ? supposed to be interpreted as the summation of these sequential barrier potentials as distance increases?

Penazzi et al [59], who introduced the double-barrier approximation, say

“As the PE molecule penetrates in between the CNTs at CNT−CNT separations longer than the critical distance of 6.0 Å, the resonance between the tails of the CNT wave functions and the frontier orbitals in the PE molecule located in between the CNTs is enhancing the tunnelling”, see fig 5 in [59]. We interpret it as that for a charge it is easier to be transferred across polymer than across vacuum.

This explanation is added in the revision.

12. 159 This sentence is concerning. If polymer molecules were able to enter the gap, would that not increase the tunneling barrier? Or is there some other physics going on here that acts to reduce the tunneling barrier?

Please see the previous answer. This is what happens in the result of electrical molecular dynamics simulations of Penazzi et al. which are accepted in literature as input for RVE simulations. We agree with the point of view that intuitively the presence of polymer as insulator is expected to increase the potential barrier. But Penazzi is talking about tunnelling through polymer electron orbitals and sequential tunnelling from CNT to orbital to CNT which is easier than tunnelling through vacuum at a longer distance.

13. Figure 1b: If Δ? is different with d < 0.6 and d > 0.6, wouldn’t the line slopes be appreciably different before and after d=0.6nm for all cases?

The slopes are different. This can be seen if the lines are prolonged to the left. They will intersect at the left edge of the graph.

14. 176 Is this relationship between Δ?2 and Δ?1 typical? What about the selection of polymer cutoff distance as 0.6nm? Is there any further justification for these choices?

The two-values approximation for the tunnelling conductivity was proposed in [59], who calculated (atomistic modelling) the polymer cutoff distance as 0.6 nm. This value is assumed in the present calculations for epoxy in the absence of better estimations. It was used as such in several publications.

This justification and further references are added in the revision.

15. 223 This is a slightly more detailed explanation than when the concept was introduced in the earlier section. Should have the more detailed explanation in the earlier section.

More detailed explanation is provided, see answers to remarks 11-14

16. 250 What is the original threshold for the maximum distance at which contact points are identified? If there is one, would there not be new contact pairs in the deformed configuration? The reverse may also be true, with pairs exiting out of range for conduction. Are there scenarios accounted for in the model, and if not, can the authors discuss the potential consequence of including/excluding this?

The threshold of 1.4 nm is mentioned in Appendix B.

The idea of the present algorithm is that no contacts are severed and no contacts are created, as the CNTs do not move freely under the deformation. The latter situation happens in a dry fibrous assembly (e.g. buckypaper for CNTs or non-woven material without thermal bonding). We are dealing with impregnated composite, with quite limited applied strain range, and the presence of the matrix prevents free creation of new contacts.

This is, of course, an abstraction, and assumption made in the approximate model. A clarifying statement is added.

17. 759 Is this an assumption being made? If so, can the authors comment further on this and its limitations?

We added a detailed discussion.

18. Where is figure Cd?

References to the parts of Fig C are corrected in the text

19. Table 1: Has this table been discussed anywhere in the manuscript?

The reference to Table 1 is added in the text in 3.1.1.

20. 322 Where exactly is this correlation being made? It is not clear.

In Fig 3a. The clarifying text is added.

21. 334 The calculation of s is not clear. What do the authors mean by distance between the CNT centerlines s_center? Why is the contact distance taken as the maximum of either s_min or s_centr – d_CNT? Is this approach the reason why the histogram shown in figure 3b shows a large frequency bin towards the higher conductance side for both R-SW and A-MW?

Clarifications and corrections are added.

As explained in 4.1.3, first paragraph, the large frequency of high conductivity bins is caused by the fact that the overwhelming majority of the contacts, 70-90%, are characterized by the maximal  conductance, corresponding to the distance smin.

22. 344 This line is not clear. What is a “section of a CNT from one tunneling connection to the next, located farther along this CNT”? Does it plainly mean the distance between contacts?

Yes, the interpretation is correct. The phrase is edited.

23. 350 Is some part of the sentence missing here?

corrected

24. Figure 4: Is the caption correct? Where are e and d?

corrected

25. Figure 6: Is the caption correct? Where is g, h , i?

corrected

26. 443 plays a paramount role

corrected

27. 584 increasingly instead of more and more

sentence corrected

28. The conclusion in general is a bit too detailed. Please work on summarizing.

The conclusions are revised.

29. 630 This explanation should’ve been there in the results discussion, if it is not there already. The discussion about scatter also may not be necessary in the conclusion.

The mentioned phrases are deleted
